

**Potential Regional Air Quality Impacts of Cannabis Cultivation Facilities in**
**Denver, Colorado**
Chi-Tsan Wang[1], Christine Wiedinmyer[2], Kirsti Ashworth[3], Peter C Harley[4], John Ortega[5],
Quazi Z. Rasool[1], William Vizuete[1*]
[1]Department of Environmental Sciences & Engineering, University of North Carolina, Chapel Hill, NC,
USA
[2]Cooperative Institute for Research in Environmental Sciences, University of Colorado Boulder,
Boulder, CO, USA
[3]Lancaster Environment Centre, Lancaster University, UK
[4]Denver, Colorado
[5]University of California Irvine, CA, USA
[*]Corresponding author: e-mail: vizuete@unc.edu; Telephone: +1 919-966-0693; Fax: +1 919-966-7911
**Abstract**

14        The legal commercialization of cannabis for recreational and medical use has

effectively created a new and almost unregulated cultivation industry. In 2018, within the
Denver County limits, there were more than 600 registered cannabis cultivation facilities
(CCFs) for recreational and medical use, mostly housed in commercial warehouses.
Measurements have found concentrations of highly reactive terpenes from the headspace above
cannabis plants that, when released in the atmosphere, could impact air quality. Here we
developed the first emission inventory for cannabis emissions of terpenes. The range of
possible emissions from these facilities was 66-657 metric tons/year of terpenes across the state
of Colorado; half of the emissions are from Denver County. Our estimates are based on the
best available information and highlight the critical data gaps needed to reduce uncertainties.
These realizations of inventories were then used with a regulatory air quality model, developed
by the State of Colorado to predict regional ozone impacts. It was found that most of the
predicted changes occur in the vicinity of CCFs concentrated in Denver. An increase of 362
metric tons/year of terpene emissions in Denver County resulted in increases of up to 0.34 ppb
in hourly ozone concentrations during the morning and 0.67 ppb at night. Model predictions
indicate that in Denver County every 1,000 metric tons/year increase of terpenes results in 1
ppb increase in daytime hourly ozone concentrations and a maximum daily 8-hour average
(MDA8) increase of 0.3 ppb. The emission inventories developed here are highly uncertain,
but highlight the need for more detailed cannabis and CCFs data to fully understand the



possible impacts of this new industry on regional air quality.
**Keywords:** *Cannabis spp.*; emission inventory; biogenic volatile organic compound; terpene;
particulate matter; ozone; air quality

## 1.  Introduction

The rapid expansion of one of the United States' newest industries, the commercial
production and sale of recreational cannabis, was recently likened to the millennial "dot com"
boom (Borchardt, 2017). With an increasing number of states passing bills to legalize
recreational cannabis, the enterprise is set to rival all but the largest of current businesses. The
cultivation, sale, and consumption of recreational cannabis annual sales revenues had reached
$1.5 billion in the US state of Colorado by 2017 (CDOR, 2018b), exceeding revenues
generated by grain farming in the state. The commercial cultivation and sale of cannabis is not
subject to the same strict environmental monitoring and reporting procedures as other
industries of similar size. While the relaxation of laws has provided certain medicinal and
economic opportunities for the states involved, the potentially significant environmental
impact on air quality due to the production of cannabis has largely been ignored.
Previous research on the wider impacts of cannabis production has been limited due to
its federal status as an illegal or controlled substance (Crick et al., 2013; Eisenstein, 2015;
Andreae et al., 2016; Stith and Vigil, 2016). As a result of this status, most studies have focused
on the pharmacological and health effects of the psychoactive constituents of *Cannabis spp.*
(Ashton, 2001; Borgelt et al., 2013; WHO, 2016), or the societal impacts associated with the
illicit nature of the industry (IDCP, 1995; Sznitman and Zolotov, 2015; WHO, 2016). The few
assessments to date on the environmental impacts of the production of *Cannabis spp.* have
centered on the detrimental effects of outdoor cultivation on ecosystems and watersheds due
to land clearance and high water demand (Bauer et al., 2015; Carah et al., 2015; Butsic and
Brenner, 2016). Studies have also quantified the energy consumption of the industry and the
resulting greenhouse gas emissions associated with indoor cultivation (Mills, 2012). Little
attention has been paid to the possible biogenic volatile organic compounds (BVOCs) emitted
from the growing of cannabis and its impact on indoor and outdoor air quality.
The only studies that have measured the composition of gaseous emissions from
cannabis have been limited to headspace samples above the plants (Hood et al., 1973; Turner
et al., 1980; Martyny et al., 2013). These studies have shown high concentrations of VOCs



such as monoterpenes ($C_{10}H_{16}$), sesquiterpenes ($C_{15}H_{24}$), and cannabinoids. These studies also
measured thiols, a sulfur-containing compound responsible for the characteristic odor of
*Cannabis spp.* (Rice and Koziel, 2015a, b). The principle (trace) components are reported to
be: α- and β-pinene, β-myrcene, d-limonene, cis-ocimene, β-caryophyllene, β-farnesene and α-
humulene (Hood et al., 1973; Turner et al., 1980; Hillig, 2004; Fischedick et al., 2010; Martyny
et al., 2013; Marchini et al., 2014; Rice and Koziel, 2015a). The precise mix of chemical
species, however, was strongly dependent on strain and the growing conditions (Fischedick et
al., 2010). It should be noted that the pharmacologically active ingredients, e.g.,
Tetrahydrocannabinol ($\Delta^9$-THC), generally have low volatility and therefore are rarely
detected in the gas-phase (Martyny et al., 2013). Measurements in (illicit) CCFs in conjunction
with law enforcement raids in Colorado in 2012 found VOC concentrations of terpenes to be
50-100 ppb within growing rooms (Martyny et al., 2013). In these cases, the CCF operation
contained fewer than 100 plants, compared with the thousands of plants found in currently
licensed premises (CDOR, 2018a). Further, the Spokane Regional Clean Air Agency (SRCAA)
study in Washington state measured indoor VOCs in seven flowering rooms and two dry bud
rooms across four different CCFs. The average terpene concentration was $361 \pm 497$ ppb in
those facilities (Southwellb et al., 2017). These indoor measurements indicate the presence of
BVOCs, but only limited studies have actually determined the chemical profile of gases
actually emitted by the growing plants. For comparison, summertime outdoor monoterpene
concentrations in forested regions of Colorado are typically less than 4 ppb (Ortega et al.,

84     2014).

Terpenoids, such as monoterpenes ($C_{10}H_{16}$) and sesquiterpenes ($C_{15}H_{24}$), are highly
reactive compounds with atmospheric lifetimes ranging from seconds to hours (Fuentes et al.,
2000; Seinfeld and Pandis, 2006). They are primarily biogenic in origin (Fuentes et al., 2000;
Guenther et al., 2012) and their reactions alter the atmospheric oxidizing capacity, resulting in
a range of low volatility products that can partition into the aerosol phase and, depending on
the concentration of nitrogen oxides ($NO_x$), lead to the formation of ozone (Laothawornkitkul
et al., 2009; Guenther et al., 2012). Both ozone and aerosols are climate-relevant components
of the atmosphere as well as criteria air pollutants (USEPA, 2016).
In Colorado, the commercial growing of *Cannabis spp.* is restricted to secure and
locked premises, resulting in indoor operations in most counties (CDOR, 2018a). Since
legalization, the number of cannabis cultivation facilities (CCFs) has risen to 1,400 across the
state of Colorado in 2018, including more than 233 registered recreational and 375 medical



CCFs within the Denver city limits alone. In Denver, the CCFs are commonly housed in
commercial warehouses and the majority of these are located near transport links such as train
hubs and major interstate highways (CDOR, 2019; Mills, 2012). Denver and the Front Range
area are currently classified as "moderate" nonattainment of the ozone standard (USEPA,
2017). Due to that status, a federally mandated State Implementation Plan (SIP) was developed
and mutually agreed upon between the state of Colorado and the United States Environmental
Protection Agency (EPA) (CDPHE, 2009). Under the terms of the SIP, Colorado Air Quality
Control Commission (AQCC) developed regulatory models to predict reductions in ozone
precursors (CDPHE, 2009). These studies have found that ozone concentrations in Denver are
VOC-sensitive, meaning that an increase in VOC concentrations will increase ozone
production (UNC-IE and ENVIRON, 2013). The location of CCFs in a VOC sensitive region
in Denver suggests a potential emission source that may impact regional air quality (UNC-IE
and ENVIRON, 2014). This work used the best available information to produce the first
emission inventory of VOCs from CCFs in Colorado. Colorado's regulatory model was then
used to determine the extent that these emissions could impact regional air quality.





**2. Materials and Methods**
**2.1 Emission Rate calculation**
Figure 1A shows the locations of the licensed 739 recreational and 733 medical CCFs
in Colorado as of March 2018 (CDOR, 2018a). Eq. (1) was first used to estimate an emission
rate for each CCF, and then all CCFs were used to build a bottom-up BVOC emission inventory.
$$ER_i = \sum_j EC_{ij} \times DPW_{ij} \times PC_{ij} \qquad (1)$$

Where, $ER_i$ (µg h$^{-1}$) is the total emissions rate for CCF $i$ based on the sum of emission
rates for all j cannabis strains; $EC_{ij}$ is the emission capacity (µg dwg$^{-1}$ h$^{-1}$) for cannabis strain $j$
in facility $i$, $DPW_{ij}$ is the dry plant weight per plant (g) for cannabis strain $j$, and $PC$ is the plant
count number for strain $j$ in facility $i$.
Since state legalization only occurred in 2014, and given the current federal illicit status
of *Cannabis spp.*, there is a lack of available data for the three parameters used in Eq. (1). The
following describes the assumptions made for a range of potential values of *EC, DPW*, and *PC*
given the best information available.
**2.1.1 Emission Capacity (EC)**
The only data of *EC* from a leaf enclosure measurement are of three strains namely:
Critical Mass, Lemon Wheel and Rockstar Kush, that were 45 days old (Wang et al., 2018).
This study found that at this growth stage the *EC* for total monoterpenes varied among strains:
10 µg gdw$^{-1}$ h$^{-1}$ for Critical Mass, 7 µg gdw$^{-1}$ h$^{-1}$ for Lemon Wheel, and 6 µg gdw$^{-1}$ h$^{-1}$ for
Rockstar Kush. The Department of Revenue (DOR) in Colorado has classified *Cannabis spp.*
in a CCF into four different growth stages: immature ( 0-24 days old), vegetative (25-79 days
old), flowering (80-132 days old), and at harvest (132-140 days old) (Hartman et al., 2018a).
Wang et al. (2018) only sampled at two intervals during the vegetative stage, and it is not
known how much *EC* will change during other stages. There are over 610 other strains (Leafly,
2018) that are grown in Colorado and the EC for these strains, or their emission rates during
other growth stages are currently unknown.
CCFs that operate in Colorado will have a wide variety of strains at all four stages of
growth whose inventory may vary throughout the year. Currently, no database exists that can
provide the number of plants by strain and growth stage. Thus, it was assumed that each CCF
had plants that consisted of only one strain and at the vegetative growth stage resulting in a
single and constant *EC* for each CCF. An *EC* of 10 µg gdw$^{-1}$ h$^{-1}$ of total monoterpenes was




chosen based on the Critical Mass strain from the leaf enclosures data (Wang et al., 2018). The
plants studied by Wang et al., however, were not grown in the optimized conditions found in a
CCF and the reported $EC$s could be conservative. Given this uncertainty in $EC$, and the variety
of possible plant stages and strains, sensitivity studies were performed with $EC$s being
multiplied by a factor of 5 and 10.

### 148    2.1.2   Dry Plant Weight (DPW)

No published studies report the $DPW$ of a *Cannabis spp.* plant. Both the states of
Colorado (METRC, 2018) and Washington (LCB, 2017; Topshelfdata, 2017) track the mass
of the commercially sold portion of the plant, the "dry bud." The Colorado database, however,
is not publicly accessible and was not available for this study. In Washington, using data from
all type of facilities (outdoor and indoor) from August–October 2017, it was found that the
average dry bud mass per plant was $210 \pm 272$ g (Fig. S1A). The Washington database also
includes the "wet bud" weight defined as the mass of the bud after it was just harvested (Fig.
S1B), but prior to the 7-10 day drying process. The total waste weight, or the remaining mass
of the plant after the buds have been harvested, is also recorded. As shown in Eq. (2), the sum
of these two masses should equal the total mass of the wet plant.
$$M_{wet\ plant} = M_{wet\ buds} + M_{waste} \tag{2}$$
Where, $M_{wet\ plant}$ is the mass of the entire wet plant (g), and $M_{wet\ bud}$ is the mass of the
wet bud (g), and $M_{wet\ waste}$ is the mass of the waste (g).
Data from August-October of 2017 were used with Eq. (2), to estimate the wet plant
weight resulting in an average of $3.77 \pm 3.62$ kg (Fig. S1C). The large range in mass is due to
the different growing conditions found in CCFs, and the type of strain being grown. The ratio
of the wet and dry bud mass data from Washington was used as a surrogate to determine the
percentage of water found in the total plant material as shown in Eq. (3).
$$R_{D/W} = M_{dry\ bud} / M_{wet\ bud} \tag{3}$$
Where, $R_{D/W}$ is the ratio of the masses of the dry to wet bud, and $M_{dry\ bud}$ (g) is the
mass of the harvested buds after 7-10 days of drying (Fig. S1D).
It was assumed that the same factor could be applied to the total wet plant weight to
estimate the remaining the $DPW$ as shown in Eq. (4).
$$DPW = M_{wet\ plant} \times R_{D/W} \tag{4}$$





The average and standard deviation of *DPW* was 754 ± 723 g (Fig. S1E). For the
development of these emission inventories, a *DPW* of 750 g was assumed based on the average
from the Washington database. As a sensitivity, a *DPW* of 1,500 g was chosen for one standard
deviation range, and 2,500 g was chosen based on the upper statistical boundary as shown in
Fig. S1E.

### 2.1.3    Plant Count (PC)

Counts of all plants larger than 8 inches have been recorded by the Colorado DOR once
per month since 2014. As of June 2018, there are 1.06 million plants (Hartman et al., 2018b,
a). Therefore, 1 million was used as the base number for the emission inventory. The DOR data
only provides county-level information and does not provide details concerning the number of
plants for each CCF. Thus, the county level data and number of CCFs per county was used to
calculate an average number of plants per facility. The average plant count per CCF in Denver
County was 905, and for areas outside of the county, it was 521 plants (Table S1).
From June 2016 (826,963 of plants) to June 2018 (1,062,765 of plants), the average
yearly increase in the number of plants was 118,000. Assuming this rate remains constant, by
2025, there could be 2 million plants in the state of Colorado. Hence, a sensitivity case was
developed to account for future growth that included these numbers of plants. Finally, each
CCF must apply for a tiered permit that determines the maximum number of plants that can be
grown and is shown in supplemental Table S2 (CDOR, 2019). As a sensitivity, it was assumed
that each recreational and medical CCF would contain the maximum number of the tier one
permitted plants leading to a state-wide total of 4 million plants. The average plant counts per
CCF and total state-wide plant counts are shown in Table S1.

### 2.2      Emission Inventories for Cannabis Cultivation Facilities (CCF)

Given the large gaps in knowledge, this study will focus only on variabilities in *EC*,
*DPW*, and *PC* and will hold other parameters constant. For example, to maximize growing
conditions relative humidity, temperatures, $CO_2$ concentrations, and fertilizer usage are all
optimized and vary widely by CCF. Further, this study did not consider other processes such
as trimming, harvesting and drying buds which may also release BVOCs.
For this study, it was assumed that all CCFs operated in the same way at a temperature
of 30ºC and 1000 µmol m$^{-2}$ s$^{-1}$ of photosynthetically active radiation (PAR). In addition, it was
assumed that all emissions from the plants inside a CCF enter the atmosphere. Ventilation to



the atmosphere varies widely by the operation, and there are no current regulations or industry-
wide practices that are being used to mitigate emissions.
In total, seven scenarios of emission inventories were created to explore sensitivities in
*EC*, *DPW*, and *PC* as shown in Table 1. In scenarios 1-3, the *PC* was held to a total of 1 million
and a 750 g *DPW* was assumed. The *EC* of 10 μg gdw$^{-1}$ h$^{-1}$ as reported by Wang et al. (2018)
was used in 1_EC, with a sensitivity that multiplied that rate by a factor of 5 (scenario 2_EC),
and 10 (scenario 3_EC). The remaining scenarios in Table 1 kept the *EC* constant at 10 μg
gdw$^{-1}$ h$^{-1}$. Scenarios 4_DPW and 5_DPW explored the sensitivity of increasing *DPW*, and
scenarios 6_PC and 7_PC increased the total plant count.
**2.3    Model description and analysis tools**
**2.3.1    Model protocols and evaluation**
The Comprehensive Air Quality Model with Extensions, CAMx6.10 (Environ and
Geophysics, 2017), was used to predict ground-level ozone concentrations. The model and
protocols used in this study are based on the Western Air Quality Modeling Study (WAQS) for
2011 (Adelman et al., 2016; Environ and Geophysics, 2017). The WAQS 2011b baseline
model simulation period runs from June 15$^{th}$ to September 15$^{th}$, 2011, and is driven with
meteorological data from WRF version 3.3 for the same time period and domain. The model
was initialized using Three-State Air Quality Modeling Study standard boundary and initial
conditions (Environ and Geophysics, 2017). The model domain is a 2-way nested grid at 12
and 4 km grid cell resolutions (Fig. 1B). Anthropogenic emissions were derived from EPA
National Emission Inventory (NEI) version 2011 NEIv2 with updates for point and area
sources of oil and gas emissions in the western US. The biogenic emissions inventory was
based on the Model of Emissions of Gases and Aerosols from Nature version 2.1
(MEGANv2.1) (Guenther et al., 2012). All data and supporting documentation are publicly
available via the Intermountain West Data Warehouse (IWDW) website (WAQS, 2017).
The revision 2 of Carbon Bond 6 (CB6r2) (Ruiz and Yarwood, 2013) chemistry
mechanism was used in all model runs. This groups all monoterpenes as a single compound
species, TERP.  Thus, the total monoterpenes EC reported in Wang et al. (2018) was converted
into the TERP species. TERP undergoes oxidation reactions with the nitrate radical (NO$_3$), the
hydroxyl radical (OH), ozone (O$_3$), and singlet oxygen. It should be noted that the TERP
category includes a wide variety of monoterpenes whose reaction rate constants may vary from
TERP (k$_{298}$ = 6.77 × 10$^{-11}$ molecules cm$^{-3}$ s$^{-1}$). For example, the rate constant of β-myrcene


with OH radical (Hites and Turner, 2009) is $3.35 \times 10^{-10}$ molecules cm$^{-3}$ s$^{-1}$ ($k_{298}$), which is 4
time higher than TERP and 5.6 times faster than α-pinene (Carter, 2010).

The details of the WAQS model setup protocol (Environ and Geophysics, 2017) and

model performance (Adelman et al., 2016) can be found in IWDW website. In summary, the
model performance evaluation concluded that this simulation had met all performance goals
for both maximum daily 1-hour (MDA1) and maximum daily 8-hour average (MDA8) ozone.
In the performance review report, it was found that the WAQS model had a positive bias for
ozone simulated in a 4 km × 4 km resolution domain, when compared with EPA Air Quality
System (AQS) surface monitors (MDA1: 0.8%, MDA8: 0.9%). On days when ozone
concentrations higher than 60 ppb were measured, the model had a negative bias of −6.2% for
MDA1 and −6.3% for MDA8. The model evaluation result also noted that the model
performance was best during the spring and summer months.
**2.3.2   Process Analysis**

CAMx runs used in this analysis had the process analysis (PA) option enabled

(ENVIRON, 2013). The CAMx configuration used here produces two additional files needed
for PA: the integrated reaction rate (IRR) and integrated process rate (IPR). These files include
the rates of change in concentration of every species due to chemistry and transport for every
grid cell and timestep. Python-based Process Analysis (pyPA) and the Python Environment for
Reaction Mechanisms/Mathematics (PERMM) (Henderson et al., 2010; Henderson et al.,
2011) were then applied to post-processing the CAMx PA output. PERMM was used to
aggregate the chemical and physical process rates for selected model grid cells and layers
allowing for tracking of plumes within the planetary boundary layer (PBL).



## 3.    Results

### 3.1    Emissions Inventory

The seven scenarios were used to estimate a range of emissions of BVOCs from CCFs for the entire state of Colorado. As shown in Table 2, the base case (BC) scenario estimates 731,442 ton/year of all VOCs being emitted in Colorado, of which 47% are BVOCs. The BC scenario does not include any emissions from the cannabis industry. Table 2 also shows the seven scenarios that did include CCF emissions ranked in order of their increases in state-wide BVOC emissions. As expected the CCF BVOC emissions scaled linearly with each factor that was changed in Eq. (1). In scenario 3_EC, a 10-fold increase in the emission rate $(100$ μg gdw$^{-1}$ h$^{-1}$) resulted in a 657 metric tons/year increase. Similarly, scenario 2_EC assumes 50 μg gdw$^{-1}$ h$^{-1}$ and produces 329 metric tons/year. Scenarios 4 and 5 showed the sensitivity of terpene emissions from CCFs to variation in $DPW$ while holding $PC$ constant and an $EC$ of 10 μg gdw$^{-1}$ h$^{-1}$. It was estimated that an additional 66 ton/year of emissions were produced when a 750 g $DPW$ is assumed. This doubles to 131 metric tons/year with a $DPW$ of 1500 g and reaches 219 metric tons/year with a $DPW$ of 2500 g. Comparing scenario 1_EC with scenario 6 and 7 shows how the growth in $PC$ will impact emissions of BVOCs. In Colorado, a doubling of the $PC$ increases BVOC emissions by 131 metric tons/year in scenario 6_PC and 261 metric tons/year for the 4 million plants in scenario 7_PC. The largest increases in BVOC emissions were predicted in scenarios 3_EC and 2_EC showing that the total emission rate of BVOCs from CCFs were most sensitive to $EC$.

In March 2018, Denver County housed 41% of CCFs and 55% of all cannabis plants in Colorado (Hartman et al., 2018b). As a result, about 43% of state-wide CCF BVOC emissions occur there (Table 2). Current emission inventories of Denver County show negligible amounts of biogenic emissions accounting for only 0.1% of the total state-wide BVOC emissions. CCF emissions increased BVOC emission rates in Denver Country up to 136% in scenario 3_EC. This changes the total VOC emission rate in Denver County by up to 3.5%. Other cities in Colorado do not have as high a concentration of CCFs, and thus the relative increases were smaller as shown in Table 2.

The introduction of additional cannabis BVOC emissions into model simulations increased the predicted TERP concentrations. Figure 2 shows the maximum increase in TERP concentrations for three scenarios for Denver County over the entire 90-day simulation period. Regardless of the scenario, the largest increases in TERP occurred near the largest





concentrations of CCFs. The absolute maximum changes ranged from 0.5-5.0 ppb located at
the Elyria Swansea and Globeville neighborhoods in north-central Denver. Increases in TERP
were also predicted to the north due to the dominant wind flows in that direction throughout
the simulation period. Figure S2 shows the maximum increase in TERP concentrations for the
1_EC, 5_DPW, and 3_EC scenarios in the 4 km × 4 km domain for the entire 90-day simulation
period. As expected substantially lower increases in TERP concentrations were predicted for
other cities in Colorado: 0.26 ppb in Colorado Springs and 0.24 ppb in Pueblo. Figure 3 shows
the hourly changes in TERP concentrations across the entire 4 km × 4 km domain. The largest
increases for all scenarios occurred at night with a peak of 5 ppb at 4:00 AM local standard
time (LST). Given that the hourly emissions of terpenes from CCFs were assumed constant for
24 hours, these larger nighttime changes can be primarily ascribed to the lack of
photochemistry and a shallow nocturnal PBL. These results suggest that the increases of TERP
are highly correlated with locations of CCFs, accumulate at night, and have significant losses
during the day.
**3.2 Regional Ozone impacts**
Predicted increases in hourly ozone concentrations in excess of 0.1 ppb only occurred
when terpene emissions were in excess of 219 metric tons per year, with scenarios 4_DPW,
6_PC, and 1_EC having little impact on predicted ozone. Thus, this analysis will focus on two
scenarios, 5_DPW, and 3_EC to explore potential regional ozone impacts in the present and
future. Figure 4 shows the hourly changes in ozone concentrations across the entire 4 km × 4
km domain for these two scenarios. During the daytime, the increase in TERP emissions results
in a peak ozone increase of 0.34 ppb at 9:00 AM LST for 3_EC with only minimal changes in
5_DPW. Figure 5 shows, for Denver County and the Front Range Metropolitan Area, the
locations of the daytime (6:00 AM – 6:00 PM LST) maximum increases in hourly ozone
concentrations for all 90 days when emissions were added for scenarios 5_DPW and 3_EC.
Ozone increases for the entire 4 km × 4 km domain can be found in Fig. S3. The largest
predicted ozone concentrations occurred in Denver County with impacts of 0.11 ppb in
5_DPW, and 0.34 ppb in 3_EC as shown in Fig. 5. Both scenarios show that daytime increases
in ozone were limited to Denver County and just to the northwest, west, and southwest of
Denver County.
There were also night time variations in ozone observed for the modeling domain. In
scenario 5_DPW and 3_EC, nighttime increases were more than double the increases predicted



during the day. The largest changes in hourly ozone concentrations of 0.67 ppb occurred at 0:00 AM LST (i.e. midnight) for 3_EC. Figure 6 shows the location and magnitude of the maximum changes in hourly ozone concentrations during the night (6:00 PM – 6:00 AM LST) in 5_DPW and 3_EC. The extent of ozone increases at night are primarily to the north of Denver indicating a northern outflow. The maximum increase in hourly ozone for the whole of Colorado is shown in Fig. S3, with visibly little changes at night in other cities. These model results suggest that the additional emissions of TERP have immediate impacts on local ozone production chemistry during both the day and night, but little wider impact.

A critical metric for the attainment of the NAAQS ozone standard in Denver County is the maximum daily average 8-hour ozone concentration (MDA8). Figure 7 shows the maximum difference in MDA8 for each grid cell centered on Denver County, across the entire 90-day simulation period for the 5_DPW, and 3_EC scenarios. Maximum increases in MDA8 are 0.14 ppb for 3_EC (Fig. 7B) co-located with the maximum increases in TERP concentrations.

### 3.2.1 Ozone impact at night

The maximum hourly ozone increase of 0.67 ppb for the 3_EC scenario occurred on Thursday, July 28th, 2011, at 0:00 AM LST (i.e. midnight) near the largest concentration of CCFs (see Fig. 8). In subsequent hours the plume of ozone moved slowly to the east before being dispersed by the rise of the morning PBL at 6:00 AM LST.

To investigate the nighttime ozone increases, the PA model output was analysed to quantify the chemical and physical processes producing ozone. Plume tracking was used so that only grid cells where the increase in ozone occurred were included in our analysis, which ran from 9:00 PM LST July 27th to 6:00 AM LST on July 28th. Vertical model layers were also aggregated to follow the hourly evolution of the PBL. Figure S4 provides snapshots of the horizontal grid cells used and the vertical layers that were aggregated throughout the simulation time period. For these grid cells and layers, Fig. S5 shows the changes in final ozone concentrations compared to BC and the physical and chemical process rates that impact those concentrations. Figure S5 shows that the process that contributes most to the modelled increases in ozone concentrations is chemical production.

For the chosen vertical layers and grid cells, Table 3A shows the total loss of TERP in BC and 3_EC across the entire period. For this period, TERP consumption due to the OH reactions led to a reduction in TERP from 0.01 to 0.1 ppb, $NO_3$ reactions led to a reduction


from 0.39 to 1.58 ppb, and $O_3$ reactions led to a reduction from 0.04 to 0.2 ppb, across
scenarios. These in turn increased the production of OH radicals and total peroxyl radicals
($TRO_2$). Table 3B also showes that the OH radical and total peroxyl radicals ($TRO_2$) source
increased by 10.0% and 25.1% due to the TERP initial reactions. Ultimately, this TERP
consumption in 3_EC led to increases in NO to $NO_2$ conversions via the $TRO_2$ pathway by
44%, and reduction of ozone titration by 1 ppb (0.8%), as shown in Table 3C. Thus, the
increased ozone concentration at night is due to a decrease in ozone loss rather than an increase
in production. The TERP emission in 3_EC also resulted in increasing by 27% $NO_x$ termination
products ($NO_z$). Organic nitrate (NTR) representing ~71% of $NO_z$ product increased from 0.66
ppb to 1.6 ppb (+142%) with this increased TERP emission in 3_EC. This increase in $NO_z$
production at night results in lower NO concentrations further reducing the ozone titration.
**3.2.2   Ozone impact during the day**

The maximum daytime hourly ozone increase of 0.34 ppb occurred at 9:00 AM on

Monday, July 18[th], 2011, as shown in Fig. 9. On this day, the meteorological conditions
favoured the maximum possible production of ozone. This day featured "upslope flows" that
are a common meteorological condition linked to ozone exceedances periods (Pfister et al.,
2017). We thus chose to focus on July 18[th] to understand the daytime changes in chemistry that
occur from increased BVOC emissions. As expected, the location of predicted ozone increases
coincides with the location of the strongest terpene emissions in the domain as shown in Fig.
9A. For the daytime hours of 6:00 AM – 2:00 PM LST, the PA option was used to quantify
changes in chemical processes for the grid cells and model layers shown in Fig. S6. For these
grid cells and layers, Fig. S7 shows the changes in final ozone concentrations compared to the
base case and the physical and chemical process rates that impact those concentrations. Table
S3 sums the key chemical processes for these hours. The increases in CCF emissions resulted
in a 100% increase in OH reactions with TERP producing intermediate oxidation products and
ultimately increasing OH production by 0.6%. As a result of this oxidation chemistry, there
was an increase of 0.9% in NO to $NO_2$ conversion by $TRO_2$ pathway, ultimately leading to a
0.1% increase in ozone production.
**3.2.3   Ozone impact sensitivity**

The maximum modelled daytime hourly ozone increase due to additional CCF

emissions occurred on July 18th. Using this day multiple sensitivity simulations were
performed, where CCF emissions from Denver County were incrementally increased up to
3,800 ton/year. Figure 10 shows the increase in terpene emissions from Denver County versus



the largest daily increase in hourly ozone concentrations. Figure 10A shows a linear
relationship, indicative of a VOC limited environment, where hourly ozone concentrations are
predicted to increase by 1 ppb for every 1,000 ton/year increase in TERP emissions during the
day, and 0.85 ppb at night. Also shown is the sensitivity to the MDA8 ozone where there is a
0.30 ppb increase for every 1,000 ton/year of TERP emissions. According to projected
emission inventories provided by the state of Colorado, the ozone non-attainment area was
expected to see reductions of 26.4% of $NO_x$ and 24.6% of VOC emissions by the year 2017
(Environ, 2017). Under these reduced anthropogenic emission scenarios, Fig. 10B shows how
ozone would then respond to additional CCF TERP emissions. Figure 10B continues to show
a linear relationship, where hourly ozone concentrations are predicted to increase by 1.5 ppb
for every 1,000 ton/year increase in TERP emissions during the day, and 1.8 ppb at night. In
the future case, the MDA8 ozone increases by 0.38 ppb increase for every 1,000 ton/year of
TERP emissions. Therefore, Denver will still be VOC-limited and ozone is predicted to more
sensitive to CCF emissions of terpenes.



**4.    Conclusion**

This study provides the first VOC emission inventory for the cannabis industry in the

U.S. Given the current state of knowledge of emission rates and growing practices, there are
considerable uncertainties in the basic parameters required to build such an inventory. Using
realistic bounds on each parameter, we developed seven scenarios, which resulted in estimated
emission rates that ranged over an order of magnitude. The highest emissions occur in Denver
County, with rates ranging between 36-362 metric tons/year for the different scenarios, from a
total of 66-652 metric tons/year across Colorado as a whole.

We included these additional terpene emissions in the Comprehensive Air Quality

Model with Extensions (CAMx), the model used by the state of Colorado for regulatory
monitoring and projections. Taking the worst case (3_EC) and median scenario (5_DPW) we
consider representative of current uncertainty upper boundary and future industry expansion;
we find that these projected increases in emissions lead to maximum increases in terpene
concentrations of up to 5.0 ppb. The largest impacts were seen in locations with the highest
terpene emissions coming from CCFs, i.e. in Denver County. We further found that these
increases in terpene concentrations affected the local atmospheric chemistry and air quality
with ground-level ozone concentrations increasing by as much as 0.34 ppb during the day and
0.67 ppb at night. In general, simulated nighttime increases were higher than those during the
daytime were, and we take the nighttime of July $27^{th} – 28^{th}$ as a case study to further investigate.
By applying process analysis (PA), following the evolving plume of VOCs and ozone, we find
that the initial reactions of the additional terpenes with OH, $NO_3$ and ozone result in increased
formation of peroxyl radicals which increases the NO to $NO_2$ conversion rate; also removes
the $NO_x$ to generate more $NO_z$ product. This effectively reduces the loss of ozone by reaction
with NO, increasing the total ozone concentration.

We acknowledge, however, the considerable uncertainties that surround our projections

and call for the need for continued efforts to reduce these such that a more accurate assessment
of the regional air quality implications of this industry can be made. Future studies that include
ambient BVOC measurements are critical for comparisons with model predictions.
Additionally, in the model chemical mechanism more accurate and mechanistic representation
of terpene species is needed that can reflect the current cannabis emission composition.
Currently, the model surrogate "TERP", which represents all monoterpene species in the
mechanisms, may not represent the precise rate constant for BVOC emissions from cannabis.



Further data are urgently required regarding CCF-specific information on plant counts and
weight by strain and growth stage, coupled with information about the agronomical practices
of cannabis cultivation in CCFs. Additional measurements of emission capacities of different
cannabis strains at different growth stages are also needed. Further, the emission inventory
version is for the year 2011; it may not be suitable to estimate the ozone impacts by the CCF
industry.

We chose to focus on ozone, since Denver is a moderate non-attainment area with an

ozone State Implementation Plan (SIP) (Environ, 2017; Environ and Geophysics, 2017;
Colorado, 2018) in accordance with the EPA regulations. But assessments of the impact of
these additional terpene emissions on particulate matter ($PM_{2.5}$) is warranted given the high
secondary organic aerosol (SOA) yields of terpenes from 0.3 to 0.8 (Iinuma et al., 2009; Lee
et al., 2006; Fry et al., 2014; Slade et al., 2017). It should also be borne in mind that
investigations of indoor air quality are needed given the findings of Martyny et al. (2013) and
Southwellb et al. (2017) that indoor terpene concentrations reached 50-100 ppb in growth
rooms and 30-1,600 ppb in flowering room, likely initiating intense photochemistry under the
powerful grow lamps in use in CCFs.



**Author contribution**
Chi-Tsan Wang and Dr. William Vizuete are lead researchers in this study responsible for
research design, experiments, analyzing results and writing the manuscript. Dr. Christine
Wiedinmyer and Dr. Kirsti Ashworth are also co-head researchers, and guided the research
design, assessed model results, and contributed to writing the manuscript. Dr. John Ortega, and
Dr. Peter Harley helped in collecting data and writing the manuscript. Dr. Quazi Z. Rasool
helped to analyze model results and contributed in writing the manuscript.

**Competing interests**
The Authors declare that they have no conflict of interest.
**Acknowledgments**
We want to thank the National Center for Atmospheric Research (NCAR) Advanced Study
Program (ASP), the Atmospheric Chemistry Observations and Modeling (ACOM) Laboratory
their support. NCAR is sponsored by the National Science Foundation (NSF). We also thank
the Colorado Department of Public Health and Environment (CDPHE) and the Intermountain
West Data Warehouse (IWDW) for the model data support. Any opinions, findings conclusions
or recommendations expressed in this material do not necessarily reflect the views of the
National Center for Atmospheric Research (NCAR), the National Science Foundation (NSF),
or the Colorado Department of Public Health and Environment (CDPHE). We also thank
Kaitlin Urso, Michael Barna, David Hsu, and Grant Josenhans for their invaluable assistance.

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



**Tables**

**Table 1.** Simulation scenarios and assumed values for emission capacity (*EC*) rate, dry plant weight (*DPW*), and the plant count (*PC*) for Colorado and Denver County. The base case (BC) scenario has no cannabis emissions.

| Name | *EC* | *DPW* | *PC* | |
|------|------|-------|------|------|
|  | (ug gdw$^{-1}$ hr$^{-1}$) | (gdw plant$^{-1}$) | Colorado | Denver County |
| BC | 0 | 0 | 0 | 0 |
| 1_EC | 10 | 750 | $1.0 \times 10^6$ | $5.5 \times 10^5$ |
| 2_EC | 50 | 750 | $1.0 \times 10^6$ | $5.5 \times 10^5$ |
| 3_EC | 100 | 750 | $1.0 \times 10^6$ | $5.5 \times 10^5$ |
| 4_DPW | 10 | 1,500 | $1.0 \times 10^6$ | $5.5 \times 10^5$ |
| 5_DPW | 10 | 2,500 | $1.0 \times 10^6$ | $5.5 \times 10^5$ |
| 6_PC | 10 | 750 | $2.0 \times 10^6$ | $1.1 \times 10^6$ |
| 7_PC | 10 | 750 | $4.0 \times 10^6$ | $2.2 \times 10^6$ |



**Table 2.** The estimated BVOC and total VOC emission rates (metric tons/year) for the base case (BC) scenario. Also shown are the increases in VOC emissions for all scenarios shown in Table 1 for Colorado, Denver County, Colorado Springs, Pueblo, and Boulder. The numbers in parenthesis are the percentage increases compared with the BC scenario.

| Name | Colorado | | Denver County | | Colorado Springs | | Pueblo | | Boulder | |
|---|---|---|---|---|---|---|---|---|---|---|
| | BVOC | Total VOC | BVOC | Total VOC | BVOC | Total VOC | BVOC | Total VOC | BVOC | Total VOC |
| BC | 340,268 | 731,442 | 265 | 10,465 | 5,184 | 15,143 | 5,870 | 9,184 | 3,677 | 9,820 |
| 3_EC | 657 (+0.19%) | +0.09% | 362 (+136%) | +3.5% | 60 (+1.20%) | +0.40% | 53 (+0.90%) | +0.58% | 26 (+0.70%) | +0.26% |
| 2_EC | 329 (+0.10%) | +0.04% | 181 (+68%) | +1.7% | 30 (+0.58%) | +0.20% | 27 (+0.45%) | +0.29% | 13 (+0.35%) | +0.13% |
| 7_PC | 261 (+0.08%) | +0.04% | 116 (+44%) | +1.1% | 42 (+0.80%) | +0.27% | 22 (+0.38%) | +0.24% | 12 (+0.33%) | +0.12% |
| 5_DPW | 219 (+0.06%) | +0.03% | 121 (+45%) | +1.2% | 20 (+0.39%) | +0.13% | 18 (+0.30%) | +0.19% | 9 (+0.23%) | +0.09% |
| 4_DPW | 131 (+0.04%) | +0.02% | 72 (+27%) | +0.69% | 12 (+0.23%) | +0.08% | 11 (+0.18%) | +0.12% | 5 (+0.14%) | +0.05% |
| 6_PC | 131 (+0.04%) | +0.02% | 72 (+27%) | +0.69% | 12 (+0.12%) | +0.08% | 11 (+0.18%) | +0.12% | 5 (+0.14%) | +0.05% |
| 1_EC | 66 (+0.02%) | +0.01% | 36 (+14%) | +0.35% | 6 (+0.12%) | +0.04% | 5 (+0.09%) | +0.06% | 3 (+0.07%) | +0.03% |





**Table 3.** All data summed from July 27th, 9:00 PM LST to July 28th, 5:00 AM LST for grid
cells and layers shown in Fig. S4. The base case (BC) scenario column shows the absolute
predicted values and, the subsequent columns show the predicted changes due to emissions
from the 3_EC scenario. Percentages in parenthesis are the changes in 3_EC relative to BC.
Shown are the **(A)** total amount of VOC and TERP consumed due to oxidation (ppb), the **(B)**
total amount of hydroxyl radical (OH) and total peroxyl radicals (TRO$_2$) that were generated
and their sources (ppb), and the **(C)** total amount of Nitrogen Dioxide (NO$_2$) and NOx
termination products (NO$_z$) produced and their sources (ppb).
A

|  | BC | 3_EC |
|---|---|---|
| VOC + OH | 1.36 | 1.68 (+23.5%) |
| TERP + OH | 0.01 | 0.10 (+900%) |
| VOC + NO$_3$ | 0.91 | 2.05 (+125%) |
| TERP + NO$_3$ | 0.39 | 1.58 (+305%) |
| VOC + O$_3$ | 1.80 | 1.97 (+9.40%) |
| TERP + O$_3$ | 0.04 | 0.20 (+400%) |


B

|  | BC | 3_EC |
|---|---|---|
| OH generation | 1.00 | 1.10 (+10.0%) |
| from TERP + O$_3$ | 0.03 | 0.11 (+267%) |
| TRO$_2$ generation | 34.2 | 42.8 (+25.1%) |
| from VOC initial reactions | 3.25 | 5.03 (+54.8%) |
| from TERP initial reactions | 0.47 | 1.98 (+321%) |


C

|  | BC | 3_EC |
|---|---|---|
| NO to NO$_2$ | 198 | 197 (-0.70%) |
| NO + O$_3$ | 158 | 157 (-0.80%) |
| NO + TRO$_2$ | 3.50 | 5.04 (+44.0%) |
| NO$_z$ generation | 4.91 | 6.24 (+27.1%) |
| NTR generation | 0.66 | 1.60 (+142%) |
| PAN generation | 1.54 | 1.56 (+1.30%) |
| PANX generation | 0.54 | 0.66 (+22.2%) |
| HNO$_3$ generation | 2.17 | 2.42 (+11.5%) |




**Figures**

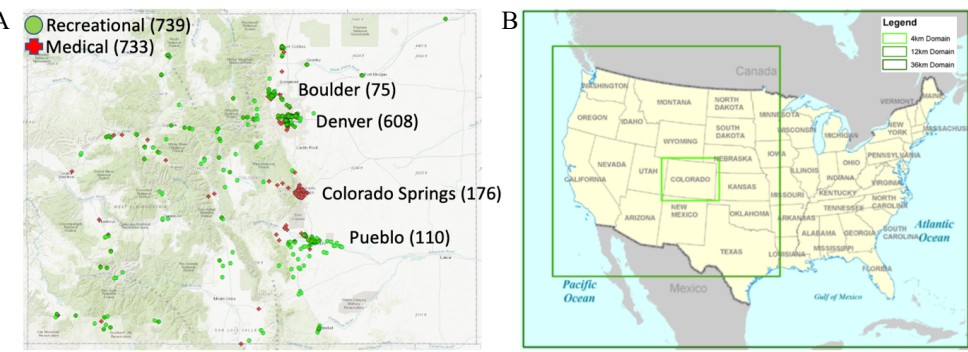


**Figure 1. (A)** The locations of medical (red) and retail (green) Cannabis cultivation facilities
(CCFs) in Colorado as of March 1, 2018. The corresponding values are the number of CCFs
found within each city. **(B)** The 36km × 36km resolution of Western Air Quality Model Study
(WAQS) and nested inner 12km × 12km resolution domains and 4km × 4km resolution domain
used by the Comprehensive Air Quality Model with Extensions (CAMx).





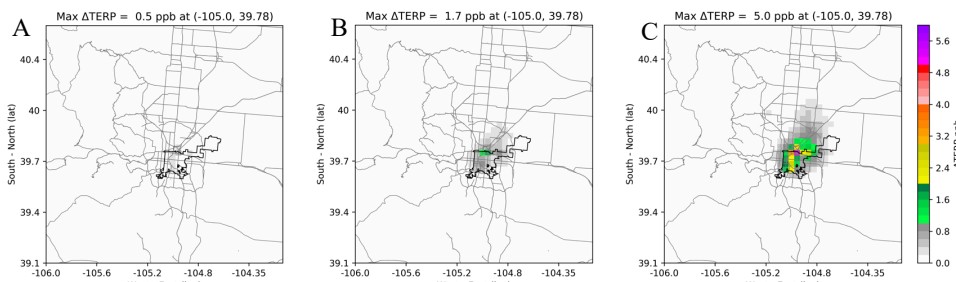

**Figure 2.** The maximum increase in TERP concentrations (ppb) for Denver County and Front Range over the entire 90-day simulation for the **(A)** 1_EC, **(B)** 5_DPW, and **(C)** 3_EC scenarios. The black outlines Denver County and the grey lines are state and interstate highways.

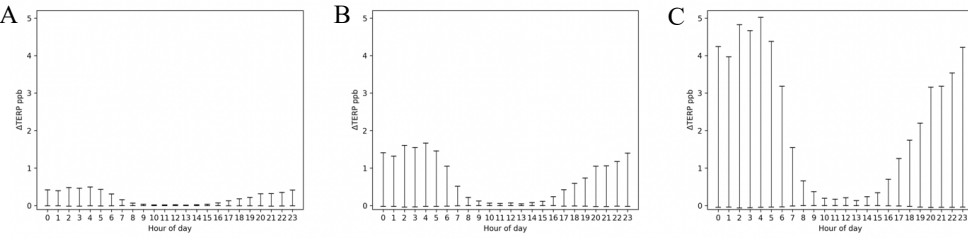

**Figure 3.** The hourly changes in TERP concentrations across the entire 4 km × 4 km domain, over the 90 days simulation for the **(A)** 1_EC, **(B)** 5_DPW and **(C)** 3_EC scenarios.





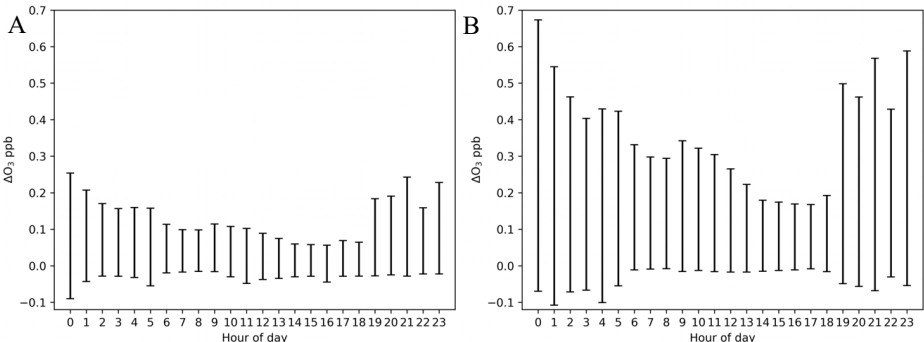

**Figure 4.** The predicted differences in hourly ozone concentrations (ppb) across the entire Colorado domain, over the 90 days simulation for the **(A)** 5_DPW and **(B)** 3_EC scenarios.

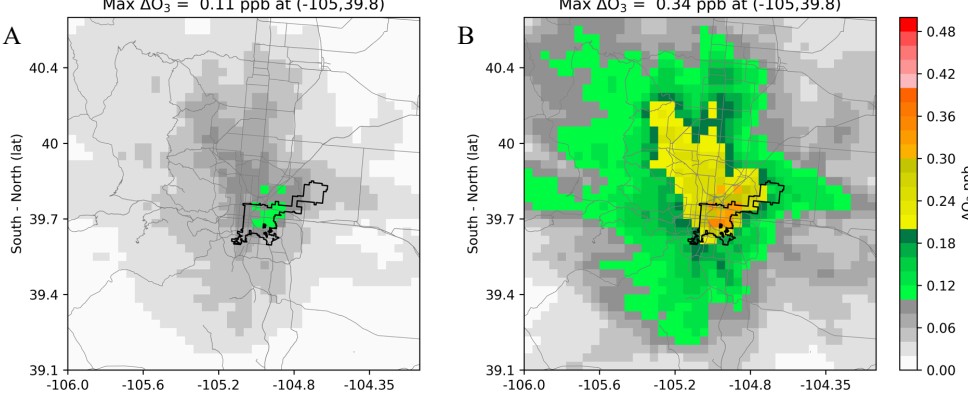

**Figure 5.** The predicted changes in hourly ozone concentrations for the Denver region from 6 AM – 6 PM LST for all 90 days of the simulation for the **(A)** 5_DPW and **(B)** 3_EC scenarios. The grey lines indicate major highways and the black line outlines Denver County.



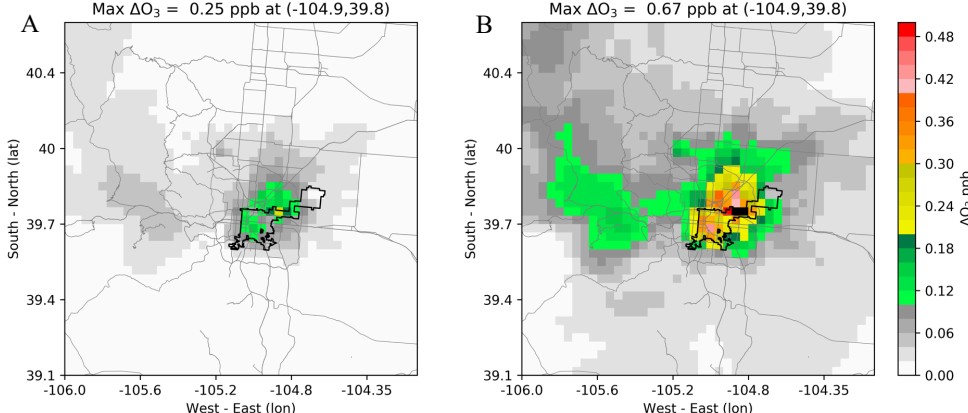

**Figure 6.** The predicted changes in hourly ozone concentrations for the Denver region from 6 PM – 6 AM LST for all 90 days of the simulation for the **(A)** 5_DPW and **(B)** 3_EC scenarios. Black regions within the map indicate ozone increase values greater than 0.5 ppb. The grey lines indicate major highways and the black line outlines Denver County.

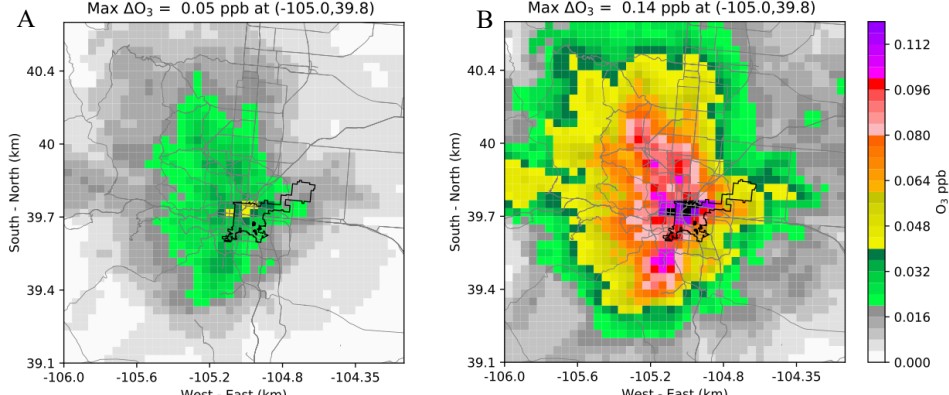

719

**Figure 7.** The predicted maximum increases in the maximum daily average 8-hour (MDA8)
ozone concentration (ppb) for the **(A)** 5_DPW and **(B)** 3_EC scenarios for the Denver region
over the 90-day simulation period. The black indicates ozone increase values greater than
0.12 ppb.

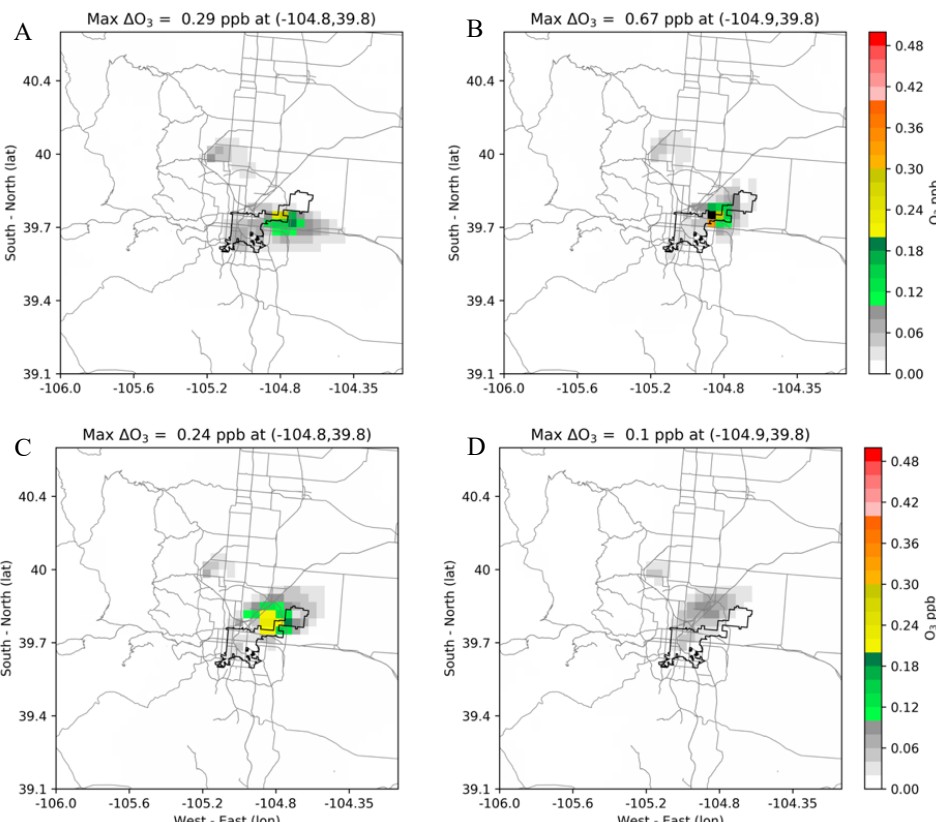

724

**Figure 8.** For the 3_EC scenario on July 28th, 2011, the largest hourly predicted ground level

ozone increases at **(A)** July 27th, 9 PM LST, and for July 28th, at **(B)** 0 AM LST (i.e. midnight),

**(C)** 3 AM LST and **(D)** 6 AM LST.



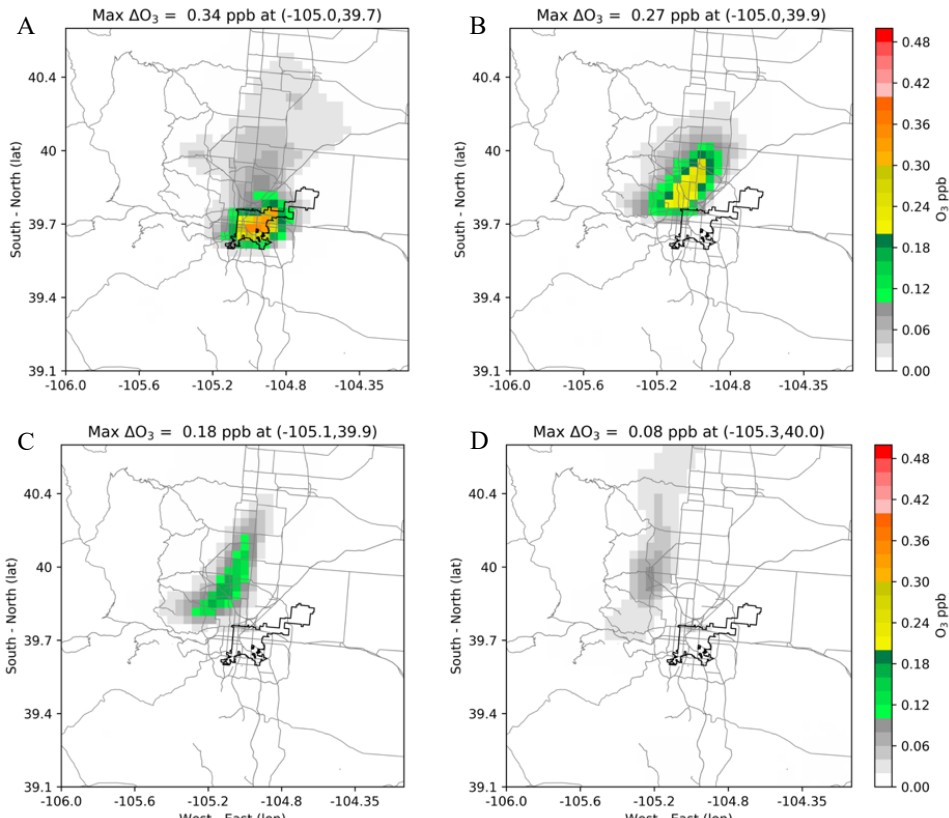

**Figure 9.** For the 3_EC scenario on July 18th, 2011 the largest hourly predicted ground level ozone increases at **(A)** 9 AM LST, **(B)** 12 PM LST (i.e. noon), **(C)** 2 PM LST, and **(D)** 5 PM LST. The maximum of 0.34 ppb occurred at 9 AM LST.



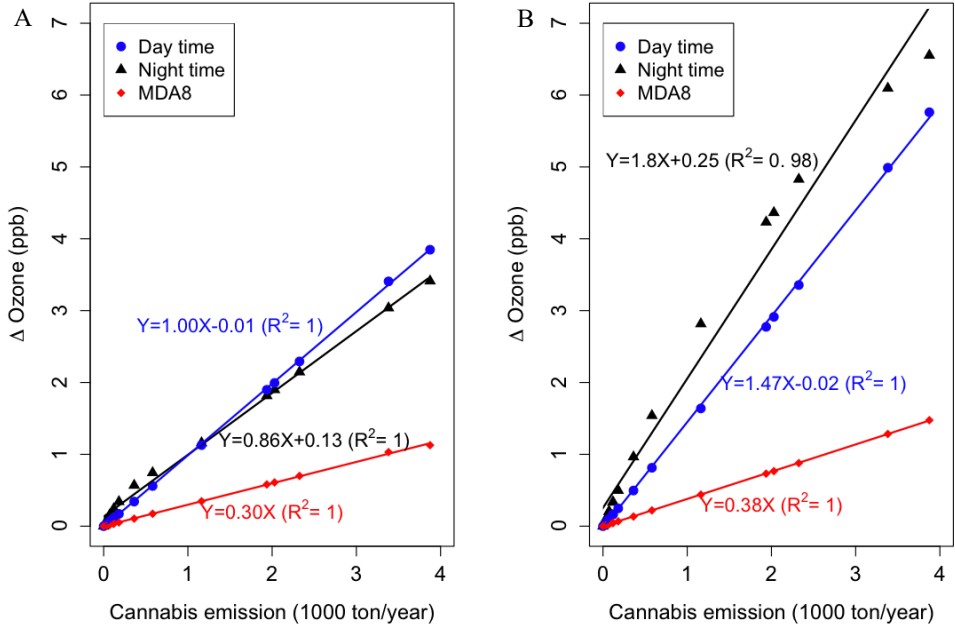

732

**Figure 10.** For July 18th during **(A)** 2011 and **(B)** 2017 the predicted maximum increase in
hourly ozone concentrations during daytime hours (6 AM – 6 PM LST) in blue, and nighttime
hours (6 PM– 6 AM LST) in black versus additional terpene emissions in Denver County. Also
shown is the response in maximum daily average 8-hour ozone concentration (MDA8) in red.