# Peer review of "Denver, Colorado"

_Atmospheric Chemistry and Physics, 2019_

## Referee Comment (RC1) · Anonymous Referee #1 · 2 Jul 2019

This manuscript is well written and provides an important initial step toward estimating VOC emissions from a growing industrial sector and demonstrates potential air quality impacts that might be anticipated given some of the range in uncertainty related to quantifying these emissions.

I only have a few minor suggestions/potential revisions.

Section 3.2.1 is rather hard to follow. The paragraph starting at line 351 talks about reductions in pollutants but the levels that follow seem to increase. Also, I found this section a little hard to follow since I am working with the presumption there are no photochemical reactions happening and everything being discussed relates to non-photochemical reactions in the model. Is that correct? I was a little surprised that hydroxyl radical production could increase overnight.

[Figure]

Section 3.2.3 line 397 discusses the increase in overnight O3 but I am not sure this would really be relevant in a discussion about implications with policy relevance since overnight O3 levels are usually well below the level of the standard.

In the conclusion section, line 402-403 needs a qualification that the emissions is the first inventory for Colorado and not the United States.
* * *

---

## Referee Comment (RC2) · Anonymous Referee #2 · 6 Aug 2019

This manuscript presents a first attempt at compiling state-wide (Colorado) emissions inventory for monoterpenes from cannabis cultivation facilities (CCFs). The new emissions inventory is incorporated into a chemical transport model to evaluate the impact of CCFs on ambient ozone concentrations. The manuscript is well written and the topic is of interest to the ACP research community and the general public, as it is important to know how much CCFs can impact air quality and provide information to decision maker on whether mitigations may be necessary to reduce the impact. Given the interest in the topic, the large gap in data and information, and generally appropriate methodology and analysis, the manuscript is acceptable for publication provided some revisions are made to clarify some points and to not overstate the results.

Because of the large uncertainties in the emissions, the study carried out sensitivity

simulations, with emissions spanning a factor of 10, to evaluate the range of potential impacts on ozone. The manuscript states that the study used "realistic bounds on each parameter" for the emissions parameterization, but it does not clearly explain why the factors chosen were considered realistic. For all parameters (EC, DPW, and PC), insufficient justification was provided on why parameter values based on leaf enclosures data of Wang et al., 2018 are considered lower bounds. The statement "...plants studied by Wang et al., however, were not grown in the optimized conditions found in a CCF and the reported ECs could be conservative" needs support/citations. Optimal growth conditions are not necessarily correlated with magnitude of monoterpene emissions. Even if one considers EC values of Wang et al., 2018 be to lower bounds, what is the basis to say that a multiple of 10 is realistic?

Even in the sensitivity case with a factor of 10 increase in emissions, the impact of increased monoterpenes associated with CCF is less than 0.5 ppb in hourly ozone during the daytime and only $\sim$ 0.14 for maximum daily average 8-hour (MDA8) ozone. This is unsurprising because the percent increase in VOC emissions is only $\sim$3.5% for Denver County for the sensitive case that has 10x the base-case CCF emissions (1_EC). Figure 10's axis going up to 4000 ton/year is hardly meaningful as 1000 ton/year increase in Denver is nearly 30 times the base-case CCF emissions, and even then the increase is only $\sim$0.38 ppb in MDA8 ozone. Thus, "further data are urgently required regarding CCF-specific information on plant counts..." is overstating the urgency of needing to improve quantification of CCF terpene emissions with respective to ambient ozone.

There are 7 sensitivity simulations listed, but in reality there are only 6 sensitivity cases because simulation 6_PC is the same as simulation 4_DPW. Because the values of EC, DPW, and PC are assumed to be constants, the emission increase is uniform across the simulation domain such that: 2_EC = 5 x 1_EC, 3_EC = 10 x 1_EC, 4_DPW = 6_PC = 2 x 1_EC, 5_DPW = 3.33 x 1_EC, 7_PC = 4 x 1_EC. Really only 3 sensitive simulations (2x, 5x, and 10 x 1_EC) was needed to cover the emissions range explored by the 7 sensitivities simulations.

---

## Author Comment (AC1) · 16 Sep 2019

**Author's Comments on manuscript ACP-2019-479 "Potential Regional Air Quality Impacts of Cannabis Cultivation Facilities in Denver, Colorado"**

Chi-Tsan Wang[1], Christine Wiedinmyer[2], Kirsti Ashworth[3], Peter C Harley, John Ortega, Quazi Z. Rasool[1], William Vizuete[1*]

[1]Department of Environmental Sciences & Engineering, University of North Carolina, Chapel Hill, NC, USA

[2]Cooperative Institute for Research in Environmental Sciences, University of Colorado Boulder, Boulder, CO, USA

[3]Lancaster Environment Centre, Lancaster University, UK

[*]Corresponding author: e-mail: vizuete@unc.edu; Telephone: +1 919-966-0693; Fax: +1 919-966-7911

**Response to our 2 anonymous reviewers**

We thank our reviewers for their comments regarding the importance and timeliness of our study. The reviewers' comments are in *grey italics* and our response is given in black.

**Anonymous referee #1:**

*This manuscript is well written and provides an important initial step toward estimating VOC emissions from a growing industrial sector and demonstrates potential air quality impacts that might be anticipated given some of the range in uncertainty related to quantifying these emissions.*

*I only have a few minor suggestions/potential revisions.*

*Section 3.2.1 is rather hard to follow. The paragraph starting at line 351 talks about reductions in pollutants but the levels that follow seem to increase. Also, I found this section a little hard to follow since I am working with the presumption there are no photochemical reactions happening and everything being discussed relates to non- photochemical reactions in the model. Is that correct? I was a little surprised that hydroxyl radical production could increase overnight.*

Section 3.2.1: We have modified the text to clarify the PA process. L341-350 now read:

"To better understand why ozone increased at night, the PA model output was analyzed to quantify the chemical and physical processes producing ozone. Plume tracking was used so that only grid cells where the increase in ozone (i.e. the plume) occurred were included in our analysis, which ran from July 27th, 9:00 PM to July 28th, 6:00 AM LST. The number of vertical model layers included in the analysis also varied to incorporate the hourly evolution of the PBL. Figure S4 provides snapshots of the horizontal grid cells used and the vertical layers that were aggregated throughout the simulation time period. Fig. S5 shows the changes in final ozone concentrations (compared to the base case) for the grid cells and vertical layers included in the analysis, as well as the physical and chemical process rates that account for these changes. Figure S5 shows that the process most responsible for increases in ozone concentrations was chemical production."

We have also expanded our explanation of the chemistry involved as the reviewer is correct that there is no photochemistry occurring at night in the model. At night, $HO_2$ radicals are produced from the reactions of VOCs with nitrate radicals ($NO_3$), OH radicals and $O_3$. OH radicals are formed when $O_3$ reacts with alkenes. We have modified the text to clarify this:

"For the chosen vertical layers and grid cells Table 3A shows the total rate of the oxidation reactions with TERP across the entire period. Throughout this time, the additional TERP emissions lead to an increase in the number of oxidation reactions thereby generating more secondary VOC products and radical species. The chemical losses of TERP increased due to reactions with: OH (from 0.01 ppb to 0.1 ppb; +900%), nitrate radical ($NO_3$) (from 0.39 ppb to 1.58 ppb; +305%), and $O_3$ (from 0.04 ppb to 0.2 ppb; +400%). Further analysis confirms that night-time oxidation chemistry leading to changes in ozone concentration are driven by $NO_3$. In the 3_EC scenario, TERP emissions only increased the annual VOC emission in Denver County by 3.5%, but this is sufficient to increase the VOC + $NO_3$ reaction rates by 125%. These increases produce more peroxyl radicals ($TRO_2=HO_2 + RO_2$) driving further oxidation and further radical production. Table 3B also shows that the generation of OH radicals from reactions of TERP with $O_3$ increased by 267%. Ultimately, these increases in initial TERP reactions with $NO_3$ and $O_3$ increase the NO to $NO_2$ conversions via the $TRO_2$ pathway by 44%, reducing the availability of NO to react with $O_3$. Thus, the increased ozone concentration predicted at night is actually due to the 1 ppb (0.8%) reduction in the loss of ozone to reactions with NO rather than an increase in actual production of ozone (Table 3C). The increased TERP emissions also increase production of $NO_x$ termination products ($NO_z$) by 27% with organic nitrate (NTR; representing ~71% of this $NO_z$ product) increasing from 0.66 ppb to 1.6 ppb (+142%). This increase in $NO_z$ production at night also results in lower NO concentrations and thus lower ozone titration."

*Section 3.2.3 line 397 discusses the increase in overnight O3 but I am not sure this would really be relevant in a discussion about implications with policy relevance since overnight O3 levels are usually well below the level of the standard.*

We agree with the author concerning the policy relevancy. It is important to note that during the vegetative stage (roughly half the growth cycle of *Cannabis spp.*) CCFs are under lights 24-hours per day so nighttime monoterpene emission rates remain similar to those during the day. Thus, these emissions can affect night-time chemistry in a way that is unique for BVOCs, as was shown in our model simulations. We have modified the text to clarify that our goal for this part of the analysis was the scientific interest of their impact on nighttime chemistry.

*In the conclusion section, line 402-403 needs a qualification that the emissions is the first inventory for Colorado and not the United States.*

To our knowledge this is the first inventory of terpene emissions for the cannabis industry to have been conducted anywhere. Our statement is ambiguous and L402-3 has been amended to read:

"This study provides the first VOC emission inventory to be compiled for the cannabis industry in Colorado, the first time such analysis has been conducted anywhere in the USA."

**Anonymous referee #2:**

*This manuscript presents a first attempt at compiling state-wide (Colorado) emissions inventory for monoterpenes from cannabis cultivation facilities (CCFs). The new emissions inventory is incorporated into a chemical transport model to evaluate the impact of CCFs on ambient ozone concentrations. The manuscript is well written and the topic is of interest to the ACP research community and the general public, as it is important to know how much CCFs can impact air quality and provide information to decision maker on whether mitigations may be necessary to reduce the impact. Given the interest in the topic, the large gap in data and information, and generally appropriate methodology and analysis, the manuscript is acceptable for publication provided some revisions are made to clarify some points and to not overstate the results.*

*Because of the large uncertainties in the emissions, the study carried out sensitivity simulations, with emissions spanning a factor of 10, to evaluate the range of potential impacts on ozone. The manuscript states that the study used "realistic bounds on each parameter" for the emissions parameterization, but it does not clearly explain why the factors chosen were considered realistic. For all parameters (EC, DPW, and PC), in- sufficient justification was provided on why parameter values based on leaf enclosures data of Wang et al., 2018 are considered lower bounds. The statement ". . .plants stud- ied by Wang et al., however, were not grown in the optimized conditions found in a CCF and the reported ECs could be conservative" needs support/citations. Optimal growth conditions are not necessarily correlated with magnitude of monoterpene emissions. Even if one considers EC values of Wang et al., 2018 be to lower bounds, what is the basis to say that a multiple of 10 is realistic?*

We thank our reviewer for their constructive comments on our sensitivity simulations. As the reviewer themselves points out, there are considerable uncertainties around all of the factors included in our estimated inventory. Here, we further clarify and justify our parameter choices in response and have modified the manuscript accordingly.

*"For all parameters (EC, DPW, and PC), insufficient justification was provided on why parameter values based on leaf enclosure data of Wang et al., 2018 are considered lower bounds."*

The DPW (plant dry weight) is not based on the enclosure measurement study. Instead, these are based on figures from Washington State Liquor and Cannabis Board and from Colorado Department of Revenue who oversee the licensing of Cannabis Cultivation Facilities.

Figure S1 summarises the available (reliable) data on material harvested from CCFs; here from Washington State. We use the yields of wet and dry buds (the marketable material) to deduce the water content in Cannabis buds and then assume that the water content is the same in the remaining plant material. From this we are able to estimate the dry weight of an average *Cannabis spp.* plant as described in the main text in L148-169. We use this mean of ~750 g (N = 18,257) as the base case value for DPW. The standard deviation in this estimate is of similar magnitude and we take the mean +1 s.d. (1500 g) as the value of DPW for our first sensitivity test. Our final value (2500 g) is considerably higher and represents the maximum yield recorded by Washington State Liquor and Cannabis Board. As the total plant count and reported yields are 3 and 4 higher respectively in Colorado than Washington state (LCB, 2017; Topshelfdata, 2017; Hartman et al., 2018b), we used this maximum on the assumption that *Cannabis spp.* cultivated in CCFs in Colorado in summer season is grown under more optimal conditions than those grown in Washington State resulting in considerably higher yields.

We have clarified this a little further in the main text. L69-174 have been modified to read:

"The average and standard deviation of DPW was 754 ± 723g (Fig. S1E). For the development of these emission inventories, a base value of 750 g was assumed for DPW based on the average calculated from the Washington database. As a sensitivity test, a DPW of 1,500 g representing the mean plus one standard deviation range was chosen. Finally, a DPW of 2,500 g, the maximum yield recorded by Washington State Liquor and Cannabis Board, was taken as the upper statistical boundary as shown in Fig. S1E. As the total plant count and reported yields are 3 and 4 higher respectively in Colorado than Washington state (LCB, 2017; Topshelfdata, 2017; Hartman et al., 2018b), we took this maximum on the assumption that *Cannabis spp.* cultivated in CCFs in Colorado in summer season is grown under more optimal conditions than those grown in Washington State resulting in considerably higher yields."

Table S2 shows the maximum number of plants permitted in a CCF for each licence tier in Colorado. This shows that our choice of value for PC is well below the maximum for Tier 1 premises. As explained in the main text (L176-181) our base value of PC is based on the current (June 2018) 1 million "mature" Cannabis plants under cultivation in Denver County, with two sensitivity simulations exploring a doubling in plant numbers (commensurate with continued expansion at the same rate, as explained in the main text in L182-185) and finally, a simulation with each CCF containing the maximum possible number of plants under a Tier 1 licence. These are summarised in Table S1.

We have changed the main text to clarify our parameter values. L175-190 now reads:

"Counts of all plants larger than 8 inches have been recorded by the Colorado DOR on a monthly basis since 2014. As of June 2018, there are a total of 1.06 million plants (Hartman et al., 2018a, b). We therefore used 1 million as the base number for the emission inventory. The DOR data only provides county-level information rather than actual number of plants per CCF. The plants were then distributed equally among the CCFs to calculate an average of 905 plants per facility in Denver County and 521 outside of the county.

Two sensitivity simulations were conducted based on the assumption that the cannabis industry in Colorado will continue to expand at similar rates in the future. From June 2016 to June 2018 the total number of plants recorded by DOR grew from 826,963 to 1,062,765, an annual average increase of 118,000. Assuming this rate of expansion remains constant, there would be 2 million plants in the state of Colorado by 2025 and this value was used in simulation 6_PC. It was assumed in simulation 7_PC that growth would accelerate in the future to the point at which each recreational and medical CCF would contain the maximum number of plants permitted under a Tier 1 license leading to a state-wide total of nearly 4 million plants. The maximum number of plants that can be grown under each licensing tier is shown in supplemental Table S2 (CDOR, 2019). The average plant count per CCF for each PC sensitivity simulation are shown in Table S1."

The monoterpene emission capacity (EC) was based on the enclosure measurements described by Wang et al., 2018. In this study, emission rates from 4 cultivars were found to vary widely among Cannabis spp. strains and across growth stages. The base value for EC was taken as 10 µg gdw$^{-1}$ h$^{-1}$ based on the average emissions from Critical Mass at the vegetative growth stage. It has been reported that during the flowering stage the bud tissues contain a significant amount of monoterpenes. Further, the Spokane Regional Clean Air Agency (SRCAA) and Washington State University measured monoterpene concentrations from indoor cannabis facilities in grow rooms (Southwellb et al., 2017). They found concentrations of monoterpenes in grow room

with 80 days old plants (1,660 ppb) to be 10 times higher than the 48 days old plants (150 ppb) suggesting that the emission rate from plants in the flowering state is higher than those measured at the vegetative stage. Since no studies in which emission rates of monoterpenes from buds have been reported, we feel that the range proposed is sufficiently wide to provide useful information regarding possible impact of this uncertainty.

We have modified the main text to incorporate these points. L130-150 now read:

"Wang et al. (2018) only sampled during the vegetative stage, and to our knowledge emission rates of monoterpenes from buds or flowers do not exist. It is not known how much EC will change during these different growth stages, but the grey literature does report that CCFs actively select cultivars to maximise the amount of monoterpenes found in the bud tissues. The Spokane Regional Clean Air Agency (SRCAA), in collaboration with Washington State University (Southwellb et al., 2017; Wen et al., 2017), measured monoterpenes in flowering rooms of CCFs in Washington state. They found concentrations of monoterpenes in grow room with 80 days old plants (1,660 ppb) to be >10 times higher than the 48 days old plants (150 ppb). CCFs in Colorado house a wide variety of strains at both vegetative and flowering stages of growth suggesting that the emission rate of monoterpenes from CCFs is higher than that measured from foliage by Wang et al. (2019). Currently, no database exists that can provide the number of plants by strain and growth stage. For the base case, it was assumed that each CCF grew only one strain and that all plants were at the vegetative growth stage resulting in a single and constant EC for each CCF; taken to be 10 μg gdw$^{-1}$ h$^{-1}$ of total monoterpenes based on the reported EC from the Critical Mass cultivar (Wang et al., 2019). Given the uncertainty in EC, the variety of possible plant stages and cultivars, the EC used in simulation 1_EC was multiplied by a factor of 5 and 10 in simulations 2_EC and 3_EC as a sensitivity analysis."

*Even in the sensitivity case with a factor of 10 increase in emissions, the impact of in- creased monoterpenes associated with CCF is less than 0.5 ppb in hourly ozone during the daytime and only ~ 0.14 for maximum daily average 8-hour (MDA8) ozone. This is unsurprising because the percent increase in VOC emissions is only 3.5% for Denver County for the sensitive case that has 10x the base-case CCF emissions (1_EC). Figure 10's axis going up to 4000 ton/year is hardly meaningful as 1000 ton/year increase in Denver is nearly 30 times the base-case CCF emissions, and even then the increase is only 0.38 ppb in MDA8 ozone. Thus, "further data are urgently required regarding CCF-specific information on plant counts. . ." is overstating the urgency of needing to improve quantification of CCF terpene emissions with respective to ambient ozone.*

We agree with the reviewer and have changed the language.

The statement now reads

"Further data are needed to reduce uncertainties in emission inventory estimates specifically those regarding CCF-specific information on plant counts, …"

*There are 7 sensitivity simulations listed, but in reality there are only 6 sensitivity cases because simulation 6_PC is the same as simulation 4_DPW. Because the values of EC, DPW, and PC are*

*assumed to be constants, the emission increase is uniform across the simulation domain such that: 2_EC = 5 x 1_EC, 3_EC = 10 x 1_EC, 4_DPW = 6_PC = 2 x 1_EC, 5_DPW = 3.33 x 1_EC, 7_PC = 4 x 1_EC. Really only 3 sensitive simulations (2x, 5x, and 10 x 1_EC) was needed to cover the emissions range explored by the 7 sensitivities simulations.*

Although the reviewer is correct that the 6_PC and 4_DPW sensitivity tests are effectively the same, the conception of these two scenarios were different. The 4_DPW emission assumed a heavier dry biomass, and 6_PC is for a future plant count estimate. In addition, the 5_DPW and 7_PC represent the upper bounds of DPW and PC showing their relative impacts on the inventory. Assessing the contribution of the individual factors on our emission inventory of, we concluded that it is the emission capacity (EC) of *cannabis spp.* that is the most significant and also the most uncertain.

Reference

CDOR, C. D. o. R., MED Resources and Statistics: https://www.colorado.gov/pacific/enforcement/med-resources-and-statistics, last access: 2, May, 2019, 2019.

Hartman, M., Humphreys, H., Burack, J., Lambert, K., and Martin, P.: MED 2018 Mid-Year Update, Colorado Department of Revenue, available at: https://www.colorado.gov/pacific/sites/default/files/2018%20Mid%20Year%20Update.pdf 2018a.

Hartman, M., Humphreys, H., Burack, J., Lambert, K., and Martin, P.: MED 2017 Annual Update, Colorado Department of Revenue, available at: https://www.colorado.gov/pacific/sites/default/files/MED2017AnnualUpdate.pdf 2018b.

LCB, L. a. C. B., Washington State Liquor and Cannabis Board: https://lcb.wa.gov, last access: 2 May, 2019, 2017.

Southwellb, J., Wena, M., and Jobsona, B., Thomas Spokane Regional Clean Air Agent (SRCAA) Marijuana Air Emissions Sampling & Testing Project, Inland Northwest Chapter AWMA, Washington State, Oct, 2017, 2017.

Topshelfdata, Topshelfdata: https://www.topshelfdata.com/listing/any_license/state/wa, last access: 2 May, 2019, 2017.

Wang, C.-T., Wiedinmyer, C., Ashworth, K., Harley, P. C., Ortega, J., and Vizuete, W.: Leaf enclosure measurements for determining volatile organic compound emission capacity from Cannabis spp., 10.1016/j.atmosenv.2018.10.049, 2019.

Wen, M., Southwell, J., and Jobson, B. T.: Identification of Compounds Responsible for Marijuana Growing Operation Odor Complaints in the Spokane Area, Pacific Northwest International Section 57th annual conference, Boise, Idaho, 2017.